# DOES RLHF SCALE? EXPLORING THE IMPACTS FROM DATA, MODEL, AND METHOD

## ABSTRACT

This study explores the scaling properties of Reinforcement Learning from Human Feedback (RLHF) in Large Language Models (LLMs). Although RLHF is considered an important step in post-training of LLMs, its scaling potential is still largely unknown. We systematically analyze key components in the RLHF framework—model size, data composition, and inference budget—and their impacts on performance. Our findings show that increasing data diversity and volume improves reward model performance, helping process-supervision models scale better. For policy training, more response samples per prompt boost performance initially but quickly plateau. And larger reward models offer modest gains in policy training. In addition, larger policy models benefit less from RLHF with a fixed reward model. Overall, RLHF scales less efficiently than pretraining, with diminishing returns from additional computational resources. Based on these observations, we propose strategies to optimize RLHF performance within computational limits.

## 1 INTRODUCTION

Large Language Models (LLMs) have revolutionized natural language processing by learning extensive language patterns from vast datasets. A key step in these models is reinforcement learning from human feedback (RLHF) (Ouyang et al., 2022), which helps align the model's behavior with human intentions and enhances their performance across diverse tasks such as text generation Hu et al. (2024), coding (Li et al., 2022; Zhu et al., 2024), and mathematical reasoning (Zhu et al., 2024) before deployment. This approach has been successfully applied to leading models like ChatGPT (Achiam et al., 2023), Llama (Touvron et al., 2023; Dubey et al., 2024), and Claude (Bai et al., 2022b), yielding notable improvements. RLHF improves the model's behavior by integrating external feedback, such as human preferences and answer supervision, to enhance its generation. The process begins with training a reward model using human-labeled preference data or reasoning data with correctness labels, serving as a proxy for human supervision. Then, reinforcement learning is employed to optimize the policy model through iterative feedback from the reward model.

The scaling properties of large language models, which lead to considerable performance improvements with increased computational resources (i.e., more data and larger models), have been viewed as key to the success of the current LLMs. However, the scaling properties of RLHF have been less understood, while extensive studies have been conducted for both the pretraining stage (Kaplan et al., 2020; Hoffmann et al., 2022; Du et al., 2024) and supervised fine-tuning (Yuan et al., 2023). Recently, the emergence of OpenAI-o1 (openai, 2024) has demonstrated the potential of scaling reinforcement learning for reasoning tasks, but the specific methods used to achieve this result have not been disclosed. Previous works (Ivison et al., 2024; Xu et al.), including Llama2 (Touvron et al., 2023), have attempted to explore the greater potential of RLHF, i.e., PPO (Schulman et al., 2017), compared to DPO (Rafailov et al., 2024b), and Llama3 (Dubey et al., 2024) only describe its use of DPO for optimization. They paid little attention to the scaling potential and properties of current RLHF and there is still a limited practical understanding of large-scale RLHF training. This raises questions about whether current RLHF techniques can lead to significant performance improvement like OpenAI-o1, given access to more data and computing resources, or whether we have to develop a more scalable reinforcement learning algorithm for LLMs.

In this work, we systematically investigate the key components of the current on-policy RLHF framework that can be scaled in policy model and reward model training, including model size,

Figure 1: **Left**: Performance trend with different reward models and sampled responses in RLHF training using a fixed SFT model. **Middle**: Performance gain of different sizes of policy model after RLHF. **Right**: Overview of reinforcement learning from human feedback (RLHF)

data composition, and the effect of inference budget. We explore how these components impact the performance of each model individually and the final performance in on-policy RLHF methods, specially PPO (Schulman et al., 2017) and GRPO (Shao et al., 2024). Since the performance of the policy model is heavily influenced by the preceding pretraining and supervised fine-tuning (SFT) stages, we aim to investigate two problems: (1) Given a fixed SFT model, how does scaling other factors affect the policy model through RLHF? (2) With a fixed reward model and training strategy, whether a larger policy model can benefit more from RLHF? Starting with an initial SFT model, we train more than 20 models with reward and policy model sizes of 9B, 32B, and 200B parameters across varying dataset sizes. We conduct a comprehensive study of different factors and show details of the training process and evaluation results to help us understand what scales in RLHF training. In particular, we pay more attention to reasoning tasks, which are considered more scalable in previous works (openai, 2024; Yuan et al., 2023) but also conduct experiments on general chat tasks.

**Observations.** Figure 1 highlights our key findings about the scaling property of RLHF on reasoning tasks, and our important observations are summarized in the following:

1. Sampling more responses per prompt during PPO training generally improves policy model's final performance, but the benefits plateau quickly (Cf. Figure 1a).
2. Larger reward models can effectively boost performance, but the improvement still significantly falls behind the gains in Best-of-$N$ evaluation of the reward model (Cf. Figure 3).
3. Larger policy models benefit less from RLHF when using a fixed size reward model (Cf. Figure 1b).
4. Performance improves remarkably in the early stage of policy training, but additional data yields only marginal gains despite increasing training rewards (Cf. Figure 5).
5. Increasing training data for reward model improves its performance in Best-of-$N$ evaluation, with increasing prompt diversity being more effective than increasing response diversity (Cf. Figure 6).
6. Process supervision from automated labeling yields better performance than outcome supervision on the targeted task, but struggles to generalize to other tasks (Cf. Figure 6).

Overall, our empirical observations suggest that the current RLHF framework does not scale as effectively as the pretraining stage. Increased computational resources do not consistently yield significant and observable performance improvements. This limitation may stem from inaccuracies in the learned reward model or current policy optimization strategies. Further research is deserved to unlock the full potential of reinforcement learning for post-training of LLMs.

Nonetheless, our empirical study still suggests some recipes for maximizing the benefits from increased compute within the current RLHF framework. Starting from the baseline of sampling one response per prompt, the following strategies can be considered in the order for optimizing performance as compute expands:

• Sampling more responses per prompt during policy training, around 4 or 8 for efficiency, is cost-effective for most tasks. Expanding the reward model size is cost-effective, too.
• Collect as many and diverse as possible training data for reward model training, yet moderate size and high-quality prompts for policy model training.
• Process supervision should be derived for all targeted tasks rather than only part of them.

## 2 RELATED WORK

**Reinforcement learning from human feedback for language models.** The primary motivation behind reinforcement learning from human feedback (RLHF) is to align language models with human intentions and preferences (Stiennon et al., 2020; Ouyang et al., 2022; Lee et al., 2023). Subsequent studies (Shao et al., 2024; Zhu et al., 2024) also demonstrate the effectiveness of RLHF in boosting LLMs' reasoning abilities. The general pipeline of RLHF first trains a reward model to capture human preferences, and then optimizes the policy model against it using reinforcement learning algorithms such as PPO (Schulman et al., 2017) and its variants (Ahmadian et al., 2024; Shao et al., 2024; Li et al., 2023). Despite its success in leading models like ChatGPT and Claude (Bai et al., 2022a;b), offline methods (Rafailov et al., 2024b; Ethayarajh et al., 2024; Ji et al., 2024) has become widely popular in open-source community due to its simplicity and stability, whereas PPO is resource-intensive and harder to train. Recently, increasing efforts have been made to reveal the secrets in online RLHF training and achieved encouraging performance across various tasks (Ivison et al., 2024; Xu et al.; Shao et al., 2024; Touvron et al., 2023). In particular, Ivison et al. (2024) investigates strategies for improved PPO across different aspects and demonstrates significant advantages of PPO over DPO. In this work, we focus specifically on the scaling properties of RLHF and investigate the extent to which its performance can benefit from increased compute.

**Scaling properties of language models.** Scaling is one of the key factors leading to the success of powerful language models and provides crucial insights into continuous improvement of LLMs' performance. Kaplan et al. (2020); Hoffmann et al. (2022) study the scaling laws for pretraining and demonstrate that scaling model size and training tokens can both lead to predictable improvements. Du et al. (2024) further build the connection between pretraining loss and downstream performance and claim that pretraining loss, rather than model size, is the key to predicting emergent abilities. In the context of reinforcement learning for LLMs, Gao et al. (2023); Cobbe et al. (2021) explore the scaling laws in reward modeling under a synthetic setting and show that over-optimizing proxy reward models can degrade true performance in RLHF. The same problem is also observed in direct policy optimization when scaling the training (Rafailov et al., 2024a). Recently, OpenAI-o1 (openai, 2024) has revealed the potential for scaling reinforcement learning at both training and inference time and significantly boosts the reasoning abilities of LLMs. This development emphasizes the importance of scaling reinforcement learning techniques and raises questions about whether the existing RLHF paradigm can achieve comparable scaling performance.

## 3 STUDY SETUP

We first describe the key components in reinforcement learning from human feedback (RLHF) before moving into our extensive empirical study. We follow the general RLHF framework, as shown in Figure 1(c), which first trains a reward model and then uses the model to guide policy training using reinforcement learning. But our implementation has several important differences. First, we train a unified model for both human preference and reasoning tasks using a multi-task objective instead of training multiple separate reward models. Second, during policy training, we sample multiple responses for each prompt and apply additional reward clipping and normalization, which lead to more stable policy training.

### 3.1 REWARD MODEL TRAINING

The reward model is usually trained on a preference dataset $D_P$, in which each prompt $x$ is associated with two responses, $y_c$ and $y_r$, along with a preference order (denoted as $y_c > y_r$, suggesting $y_c$ is preferred). These responses are sampled from an existing LLM and the preference generally comes from human annotation or online feedback. The reward model $R_\psi(x, y)$ is a scalar function and consists of a transformer backbone with a regression head replacing the traditional language modeling head. Training the reward model involves minimizing the following loss function:

$$\mathcal{L}_{pref} = -\mathbb{E}_{(x,y_c,y_r)\sim D_P} \left[ \log \sigma \left( R_\psi(x, y_c) - R_\psi(x, y_r) \right) \right]. \tag{1}$$

For reasoning tasks, which generally have a definitive correct answer $\mathcal{I}(y) \in \{0, 1\}$, such as math and programming, it is more effective to approach them as binary classification problems rather than pairwise ranking tasks, as shown in previous studies (Uesato et al., 2022; Lightman et al., 2023; Shao

et al., 2024). Consequently, in this work, we adopt a multi-task learning approach to train our reward model. For preference data, we optimize the model using Eq (1), while for binary-labeled data, we employ cross-entropy loss for optimization:

$$\mathcal{L}_{R(\psi)} = -\mathbb{E}_{(x,y_r,y_l) \sim D_B}[y_l * \log R_\psi(x, y_r) + (1 - y_l) * \log(1 - R_\psi(x, y_r))] + \mathcal{L}_{pref} \quad (2)$$

where $y_l \in \{0, 1\}$ indicates the correctness of the response. We adopt a similar approach to train the process reward model (PRM) with the following differences. First, we retain outcome supervision for human preference data while introducing process supervision signals only for reasoning-related tasks. Second, the cross-entropy loss is applied to all intermediate steps in addition to the final step. In this work, to produce process supervision, we follow the approach in (Wang et al., 2024; Luo et al., 2024) to annotate the intermediate steps automatically with additional inference rollouts.

**Research question:** Regarding the reward model, we aim to investigate the effects of *model size* and *data scaling*, which includes both the scale and diversity of the reward model training data, as well as *process supervision*. Since previous work (Snell et al., 2024) observe that using the PRM's prediction at the last step shows the best performance in Best-of-$N$ evaluation, it can be considered as a specialized reward model scaling (e.g., in data annotation) that incurs higher inference cost.

### 3.2 POLICY MODEL TRAINING

The objective of policy training is to maximize the reward collected by its generated responses. The training process involves four models: a reward model that provides feedback, a policy model for optimization, a reference model for regularization, and an optional critic model for stabilizing policy training. For each prompt, rather than using pre-generated responses, a response $y$ is sampled from the latest policy model $y \sim \pi_\theta(y|x)$. This response is scored by the reward model, producing a response-level reward $r = r_\psi(x, y)$. In general, a more precise reward can lead to improved policy training. In our practice, multiple responses $\mathbf{y} = \{y_1, ..., y_N\}$ are sampled for a prompt $x$ to improve the utilization of prompt data, and all of them are used for policy training. We also perform reward normalization: given the raw reward $\boldsymbol{r}_0 = \{r_i\}_{i=1}^N$ for all responses in $\mathbf{y}$, the final reward for policy optimization is obtained via $\boldsymbol{r} = \{\frac{r_i - \text{mean}(\boldsymbol{r}_0)}{\text{std}(\boldsymbol{r}_0)}\}_{i=1}^N$.

Generally, the policy training maintains a KL divergence penalty to prevent the model from moving too far away from the SFT model causing degeneration:

$$\max_{\pi_\theta} \mathbb{E}_{x \sim \mathcal{D}_\pi, y \sim \pi_\theta(y|x)} [r_\psi(x, y)] - \beta \mathbb{D}_{\text{KL}}(\pi_\theta \| \pi_{\text{ref}}) \quad (3)$$

where $\pi_{\text{ref}}$ refers the SFT policy that usually initializes policy training. Since directly optimizing Eq (3) can be unstable, policy training is often realized via PPO (Schulman et al., 2017) and its variants (Shao et al., 2024) using a clipped version of policy gradient for more conservative and stable learning. In general, these policy-gradient based methods for RLHF training can be written in the following general format:

$$\mathcal{J}(\theta) = \mathbb{E}_{q \sim P(Q), o \sim \pi_{\theta_{\text{old}}}(O|q)} \left[ \min \left( \frac{\pi_\theta(o|q)}{\pi_{\theta_{\text{old}}}(o|q)} A_t, \text{clip} \left( \frac{\pi_\theta(o|q)}{\pi_{\theta_{\text{old}}}(o|q)}, A_t, 1 - \epsilon, 1 + \epsilon \right) \right) \right] \quad (4)$$

where $\theta_{\text{old}}$ is the policy during the policy gradient steps from which the evaluated responses are sampled. $A_t$ is the advantage. $A_t$ is obtained through the combination of the reward $r_\psi(x, y)$ and a estimated value $V_\phi(x, y)$ in PPO, and $A_t$ equals to $r_t$ for Group Relative Policy Optimization (GRPO) (Shao et al., 2024) or REINFORCE.

**Asymmetric reward shrinking.** In our experiments, we found that policy-gradient methods without a critic model are less stable compared to PPO. Inspired by the issue of negative gradients in direct policy optimization (Rafailov et al., 2024b), which can result in unpredictable behavior and abnormal outputs, we apply an asymmetric shrinking technique that maps the reward in an asymmetric way:

$$r_i = \begin{cases} \alpha \cdot r_i & \text{if } r_i < 0, \\ r_i & \text{otherwise.} \end{cases}$$

where $\alpha$ is a constant and $\alpha < 1$. In our experiments, this technique contributes to more stable training and leads to a steady increase in training rewards.

**Research question:** We first examine whether larger policy models can benefit more from RLHF given a fixed reward model and training strategy. Then from the aforementioned procedures about policy training, we examine what factors can benefit RLHF when scaled up: i.e., whether one should introduce more responses per prompt or more prompts, whether a larger reward model would be beneficial, and whether different reinforcement learning algorithms matter. Given a fixed size policy model, we examine how these factors affect the final performance of the trained policy with increased computing resources.

# 4 EXPERIMENT

## 4.1 EXPERIMENT SETUP

**Data construction.** We collect diverse datasets to train both the reward and policy models. These datasets include general open-chat, math, code, and instruction-following tasks. Specifically, the collection comprises MATH (Hendrycks et al., 2021), Numina-Math (Jia LI & Polu, 2024), a Chinese K-12 dataset, ShareGPT, code contest data, and additional self-collected open-domain chat data. The response data for reward model training is sampled from a series of GLM models (GLM et al., 2024), including GLM4-9B-chat and larger models. The dataset distribution is approximately 40%, 30%, and 30% for general open-chat, math, and code, respectively.

**Training settings.** We mainly experiment with GLM4-9B (GLM et al., 2024), a widely-used open-source LLM in many recent studies. We first train an SFT model based on GLM4-9B, which is then used for reward and policy training. All models are trained with OpenRLHF (Hu et al., 2024) framework. For policy training, we use a constant learning rate, generate rollouts of $1024 \times M$ prompts, and take a gradient step per $128 \times M$ samples, where $M$ represents the number of sampled responses per prompt during training. Outcome reward models are used for all experiments. To examine the effect of model size, we also conduct experiments with larger models with around 32B and 200B parameters. Unless specified, models in policy training are based on GLM4-9B-SFT.

**Evaluation.** We conduct evaluations on diverse benchmarks, including reasoning, e.g., MATH (Hendrycks et al., 2021) and GPQA (Rein et al., 2023), and LiveCodeBench (Jain et al., 2024), general tasks, e.g., MMLU (Hendrycks et al., 2020), and alignment, e.g., AlignBench (Liu et al., 2023). We evaluate on a subset of AlignBench which excludes data from mathematical reasoning and Chinese reasoning categories to focus specially on the general preference alignment evaluation. For reward model evaluation, we follow the Best-of-N strategy and sample $N$ responses from the policy model for each prompt. These responses with the prompt are then fed into the reward model and the response of the highest-score is selected as the final output for evaluation. For policy model evaluation, we report the performance with responses generated by greedy decoding.

## 4.2 SCALING OF POLICY MODEL TRAINING

### 4.2.1 EFFECTS OF RESPONSE SAMPLING

**Setting.** To investigate the impact of response sampling during policy training, we conduct experiments using PPO, and sample 1/2/4/8/16 responses for each prompt respectively. As stated in Section 4.1, the batch size is proportional to the number of sampled responses to maintain consistent gradient update steps across different training experiments.

**Results.** The evaluation results are reported in Figure 2. Generally, increasing the number of samples leads to better performance across most tasks. This suggests that more sampled responses allow the policy to make better-informed improvements by learning from a wider variety of reward feedback. But the impact of increased sampling is not uniform across tasks. For example, the MMLU task shows an inconsistent trend, especially under the larger reward model; and in AlignBench, the performance even significantly dropped with 16 responses. This implies that the benefits of increased response sampling can be task-specific. The most substantial improvements often occur with moderately increased samples (from 1 to 4). However, the rate of improvement tends to slow down with more samples (8 to 16), suggesting potential diminishing returns. This is particularly noticeable in the MATH and LiveCodeBench tasks.

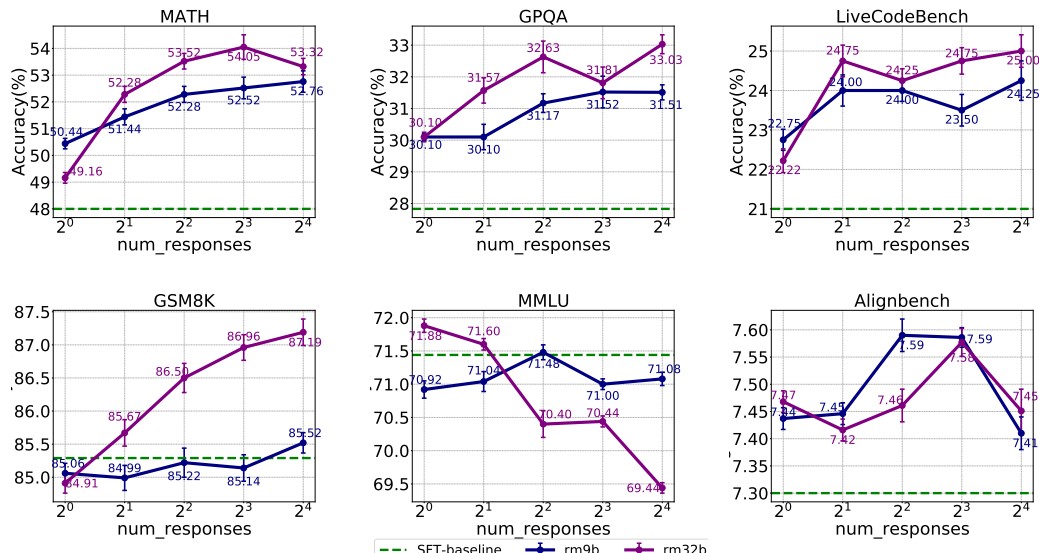

Figure 2: Results of policy model after PPO training. We report the performance using different reward model sizes and different numbers of responses sampled per prompt during training.

To investigate the relation between sampling multiple responses in policy training and Best-of-$N$ evaluation of reward models, we report the comparison results in Figure 3, which is constructed from the average performance of four reasoning-related datasets, i.e., MATH, GPQA, LiveCodeBench, and GSM8K. Significant performance improvement in Best-of-$N$ results indicate that increased response sampling tends to yield more high-quality responses and the reward model can effectively identify them, thus leading to performance improvement in policy training. However, the gain in policy training lags considerably behind the improvements in Best-of-$N$ evaluation of reward models, indicating that there is still room for further optimizing in policy.

We also investigate the factors in driving the policy update during the training process. Figure 4 (a) illustrates that more sampled responses lead to a faster reward increase during policy training, which could be attributed to a larger batch size or more accurate aggregated reward estimation.

Overall, increasing the number of sampled responses generally boosts policy model performance in most tasks and shows observable scaling trends in reasoning-related tasks. The most notable improvement occurs when increasing the sample size from 1 to 4, with relatively smaller returns as more samples are added.

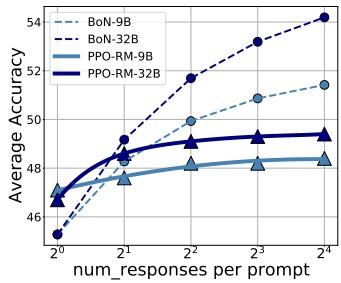

Figure 3: Results of Best-of-$N$ (reward model) and PPO training ($N$ responses), with average performance of MATH, GPQA, LiveCodeBench, and GSM8K

### 4.2.2 EFFECTS OF REWARD MODEL SIZE

**Setup.** We conduct experiments on PPO with reward models of 9B and 32B parameters. The two reward models are trained with the same data except for minimal hyper-parameter adjustments.

**Results.** Figure 1 (a) shows that larger reward models can consistently bring improved performance in reasoning-related tasks when sampling more than 2 responses during training. From Figure 2, it is observed that the 32B reward model consistently outperforms the 9B reward model in reasoning-related tasks like MATH, GPQA, GSM8K, and LiveCodeBench.

However, the performance gain is not uniform across all tasks. For MMLU, whose performance highly relies on the policy model's pretraining stage, training with a large reward model starts stronger but pays more alignment tax with increased samples. And for AlignBench, training with the smaller

Table 1: Comparison of PPO and GRPO policy training methods.

|  |  | MATH | GPQA | LiveCodeBench | GSM8K | MMLU | *Average* |
|---|---|---|---|---|---|---|---|
| SFT | Reward | 48.2 | 27.8 | 21.0 | 85.3 | 71.4 | 50.7 |
| **N_SAMPLE=4** |  |  |  |  |  |  |  |
| PPO | 9B | 51.4 | 30.1 | 24 | 85.0 | 71.0 | 52.3 |
| GPRO | 9B | 51.5 | 31.1 | 21.0 | 86.1 | 70.3 | 52.0 |
| PPO | 32B | 53.5 | 32.6 | 24.3 | 86.5 | 70.4 | 53.5 |
| GRPO | 32B | 52.8 | 32.3 | 24.3 | 86.9 | 72.0 | 53.6 |
| **N_SAMPLE=16** |  |  |  |  |  |  |  |
| PPO | 9B | 52.8 | 31.5 | 24.3 | 85.5 | 71.1 | 53.0 |
| GRPO | 9B | 52.6 | 31.7 | 22.8 | 86.1 | 72.2 | 53.1 |
| PPO | 32B | 53.3 | 33.0 | 25.0 | 87.2 | 69.4 | 53.6 |
| GRPO | 32B | 52.7 | 33.9 | 24.5 | 87.0 | 71.4 | 53.9 |


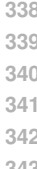

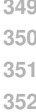

Figure 4: The training process of PPO and its variant GRPO. (a): Training rewards of PPO with different numbers of responses sampled per prompt. (b): The training reward comparison between PPO and GRPO. (c): KL-reward relation. (d): Response length change during training.

reward model even shows a clear advantage. The reason may be that the quality of learning human preferences is not scalable and a larger reward model tends to overfit the noise in the training data.

To summarize, larger reward models generally lead to better performance of the policy model in reasoning-related tasks, but the benefits are uncertain for other tasks

### 4.2.3 EFFECTS OF MORE TRAINING DATA

**Setup.** We collect more than 200k policy model training prompts and conduct experiments with a constant learning rate. The goal is to monitor the performance of policy model during training and examine whether the policy model can benefit from more training data.

**Results.** The training progress is shown in Figure 4, and the corresponding intermediate downstream evaluation performance can be found in Figure 5. The reward steadily increases as training progresses, indicating that PPO can effectively optimize toward the predefined target and maximize the reward. However, the downstream evaluation results behave differently. For most reasoning tasks, such as MATH and LiveCodeBench, performance shows rapid improvement in the early stages, but quickly plateaus, with only marginal gains in the later phase of training. Unlike the scaling trends in pretraining and supervised fine-tuning, current RLHF solution cannot benefit from more training data, and thus scaling the prompts does not lead to significant performance improvement.

Overall, we conclude that the performance evaluated by reward model during policy training is not strongly correlated with the policy model's downstream task performance. Higher rewards do not necessarily translate to better downstream performance. And the main improvement of the policy model in downstream tasks happens in the early stage of policy training.

### 4.2.4 SCALING OF POLICY MODEL SIZE

**Setup.** We investigate the performance gain of different sizes of policy models after RLHF with a fixed reward model and training strategy. The experiments include policy models ranging from 9B to 200B parameters, alongside reward models of 32B and 200B.

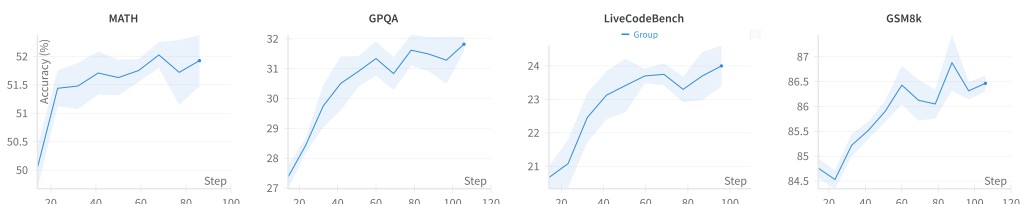

Figure 5: Downstream evaluation performance during PPO training. We report the average performance of multiple runs, e.g., different reward model sizes, and sampling strategies, on four benchmarks to explore the general trends.

**Results.** Figure 1(b) shows the average performance gain on reasoning tasks, i.e., MATH, Live-CodeBench, and GPQA. It is observed that for different reward models, the performance improvement of the policy model diminishes as its size increases. For example, when using a 32B reward model, the average performance gain consistently decreases from 4.4% to 1.9% as the policy model size grows from 9B to 200B. The results indicates that in current RLHF, larger policy models would benefit less from RLHF training, which is even inverse scaling.

### 4.2.5 TRAINING ALGORITHM: PPO V.S. GRPO

**Setup.** We also compare the two policy training algorithms—PPO and GRPO, and investigate their difference in the training behavior and final results. We apply the asymmetric reward shrinking proposed in Section 3 to stabilize the training of GRPO.

**Results.** The overall performance of policy models is shown in Table 1. We observe minimal differences between the two methods and their performance is largely similar even across different reward models and sampling strategies. One noticeable difference is that GRPO generally better maintains the performance in MMLU, yet PPO might cause a performance drop compared to the SFT baseline. Figures 4b, 4c, and 4d report the reward, KL divergence, and response length during the training process. Both algorithms demonstrate an increase in reward, and PPO exhibits better stability through a steadier reward increase. GRPO tends to significantly increase the divergence between the policy and the SFT model and also leads to a marked higher increase in response length.

Overall, while the two methods exhibit different training behavior, they achieve comparable performance in policy model evaluation and exhibit similar scaling trends.

### 4.3 REWARD MODEL

### 4.3.1 DATA DIVERSITY AND SCALE

**Setup.** We examine the impact of data scale and prompt diversity on reward model learning. We set up experiments on the math task and use the same training set in (Lightman et al.), which includes 11k questions. To vary the scale of data, we utilize all 11k questions and sample 5, 10, 20, and 40 solutions per question. To assess data diversity, we fix the number of solutions at 40 per question and progressively increase the number of questions, starting from 1/8 of the total setup to the full dataset.

**Results.** The results are listed in Figure 6 (left) presenting the Best-of-4 and Best-of-64 performance of GLM4-9B-chat on the MATH-500 dataset. Generally, increasing training prompts or solutions both benefit the performance in Best-of-$N$ evaluation. Specifically, experiments with higher prompt diversity consistently outperform those with more solutions per prompt for a given size of training set (shown in the shadow region), and the gap is more significant with more responses in the evaluation. Furthermore, the performance improvement appears nearly linear as prompt diversity increases with a fixed number of solutions per prompt. It suggests a robust and scalable benefit to expanding the set of training prompts. Therefore, to boost the performance of the reward model, the top priority is to collect diverse prompts, and then sample multiple responses, especially when resources are constrained.

Overall, reward model training shows promising data scaling trends. Increasing prompt diversity proves more effective than generating multiple responses per prompt.

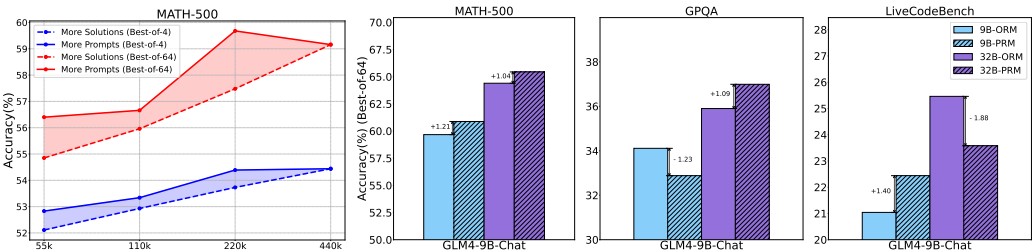

Figure 6: **Left**: Performance of reward models on MATH-500 with varying training data sizes. **Right**: Results of Best-of-64 performance with different outcome and process supervision models.

### 4.3.2 PROCESS REWARD V.S. OUTCOME REWARD

**Setup.** We evaluate the performance of process reward model (PRM) compared to outcome reward model (ORM). In PRM training, only the intermediate steps in the solutions to math-related problems are considered, as these steps can be automatically annotated by assessing the success rate of obtaining a correct answer by rolling out from each intermediate step, following in (Wang et al., 2024). For other tasks like code and human preference, only the final label is used. As a result, the trained PRM is an ORM for general tasks but is specifically tailored for math-related reasoning tasks as PRM.

**Results.** The results are illustrated in Figure 6 (right). PRM exhibits distinct behavior for different datasets. For MATH-500, PRM consistently outperforms ORM in Best-of-64 evaluations across different reward model scales. However, the advantage is not consistent for GPQA and LiveCodeBench, whose process supervision is not covered in the training data. PRM-9B performs better than ORM in LiveCodeBench but worse in GPQA, while PRM-32B shows the opposite trend. This suggests that PRM struggles to generalize to tasks outside its training data, such as GPQA. And expanding training data to cover non-math tasks could further improve PRM's performance. Considering the generalization problem, we only use ORM in previous policy training experiments.

Overall, PRM can be highly effective for the targeted task, but it struggles to effectively generalize to other tasks not included in the training set.

### 4.4 DISCUSSION AND LIMITATIONS

Based on the analysis above, we can summarize the factors that contribute to performance improvement with increased compute during RLHF training. These include: more sampled responses, a larger reward model for policy model training, more diverse prompts, and process supervision for reward model training. Note that we did not directly perform policy training under PRM due to the aforementioned generalization problem, and we have not found an effective strategy for generating process supervision across different tasks yet. However, RLHF training does not scale efficiently as pretraining, with improvements tending to saturate and provide only marginal gains beyond a certain point. A more concerning problem is that larger policy models benefit less from RLHF. The problem preventing RLHF scaling may be attributed to inaccuracies in reward modeling, which may lead to substantial noise in policy training. As a result, current RLHF methods do not scale and could not consistently benefit from increased compute during training. It is thus crucial to explore more scalable training methods for reinforcement learning of LLMs.

## 5 CONCLUSION

This study examined the scaling properties of reinforcement learning from human feedback (RLHF) in LLMs. We conducted extensive experiments to analyze the key components—model sizes, data, and algorithms. Our findings show that RLHF does not scale as effectively as pretraining or supervised fine-tuning in LLMs. Larger policy models benefit less from RLHF and gains from additional data and compute quickly become limited. Despite these limitations, we also find practical and beneficial strategies. Future work should focus on developing more scalable RLHF techniques to fully unleash its potential in boosting LLM performance.

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

## A APPENDIX

### A.1 BEST-OF-N RESULTS

We compare the best-of-$N$ performance of the outcome reward model (ORM) and process reward model (PRM) across different sizes. For each test sample, we initially sample $N$ responses from the policy model with a temperature set to 0.9. Subsequently, the reward model predicts a score for each response. The response with the highest score, as determined by the reward model, is considered the outcome of the reward model. The quality of this response is referred to as the best-of-N score. The evaluation encompasses the domains of math, i.e., MATH-500 and GSM8K, coding, i.e., LiveCodeBench, and reasoning, i.e., GPQA.

To assess the generalization capability of the two reward models, we evaluate their performance not only on GLM4-9B-chat (GLM et al., 2024) but also on LLAMA3-8B-Instruct (Dubey et al., 2024). Output distribution shift is common in policy model training due to the increasing divergence between the latest policy model and the SFT model as training progresses. Given that the reward model is trained on the SFT data distribution, it is necessary to test its ability to generalize to shifted distributions.

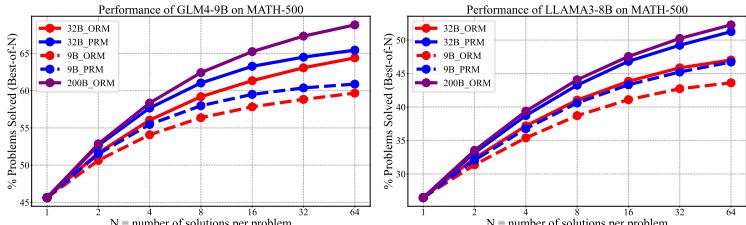

Figure 7: Best-of-N performance of ORM and PRM with different policy models on MATH-500.

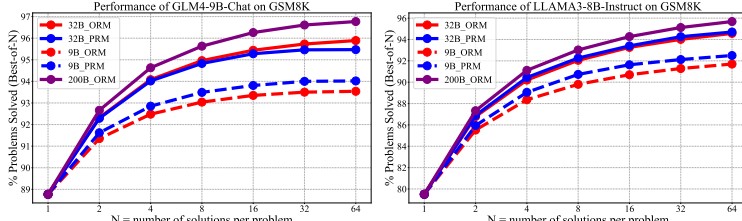

Figure 8: Best-of-N performance of ORM and PRM with different policy models on GSM8K.

The evaluation results are shown in Figure 7 for MATH-500, Figure 9 for GPQA, Figure 10 for LiveCodeBench, and Figure 8 for GSM8K. As expected, the best-of-N performance of ORM and PRM improves with the increase in model size and PRM consistently outperforms ORM in MATH. And in most cases, increasing the sampled responses leads to better performance, except for part results on GPQA and LiveCodeBench. This may indicate that the reward model is disturbed by uncertain noise and thus the decrease exists.

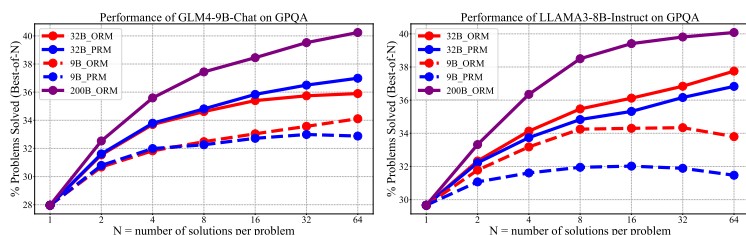

Figure 9: Best-of-N performance of ORM and PRM with different policy models on GPQA.

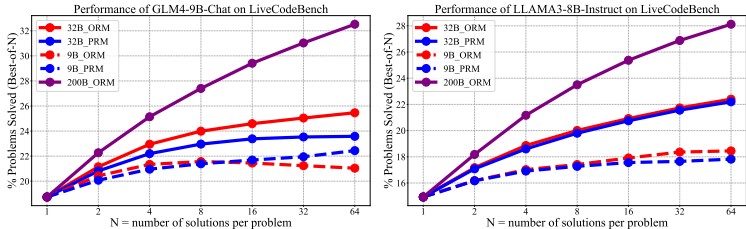

Figure 10: Best-of-N performance of ORM and PRM with different policy models on LiveCodeBench.

Comparing the results on GLM4-9B-chat and Llama3-8B-Instruct, we find that in math-related tasks, i.e., MATH=500 and GSM8K, PRM generally performs better than ORM in Llama3-8B-Instruct and thus shows better generalization ability. However, for GPQA, the PRM-9B almost leads to degenerated performance and is rather not robust. But PRM-32B shows comparable results. The observation is consistent across the two models. Therefore, we have to collect targeted process supervision data to train an effective PRM model.