# OpenReview forum: "Does RLHF Scale? Exploring the Effects of Data, Model, and Method"
_ICLR.cc/2025/Conference — Submitted to ICLR 2025_

### Official Review · Reviewer_XHYP · 2024-10-23

**Soundness:** 3
**Presentation:** 3
**Contribution:** 3
**Rating:** 8
**Confidence:** 3

**Summary:**

This paper studies how policy model performance changes as components of RLHF are scaled. Specifically they look at the effects of sampling multiple responses from the policy model for a given prompt, reward model parameter count, RLHF training example count and policy model parameter count. They also compare policy model performance when RLHF is done with PPO versus GRPO, and with process supervision versus outcome supervision.

For each component of RLHF, they plot policy model performance on a downstream task (e.g., MATH, GPQA, etc.) at different scales. Where appropriate, trends are fit to policy model performance.

The paper concludes that RLHF generally does not scale as well as pre-training, and that larger policy models do not seem to benefit as much from RLHF. Despite this, when scaled some of the components of RLHF do yield superior performance, such as sampling from the policy models multiple times, however this benefit is shown to plateau quickly.

**Strengths:**

**Originality**: To my knowledge this is the first work to directly study the scaling properties of RLHF. The studied techniques have largely appeared in the literature, but I am not aware of equivalently detailed studies of their scaling.

**Clarity**: The writing is generally clear. I did not find any part of the paper confusing. I expect Section 3 to be sufficient for someone not familiar with RLHF to read and have the necessary context for the rest of the paper.

**Quality**: The paper considers a reasonable number of datapoints for most experiments and uses well-respected benchmarks for downstream policy model performance. I think the paper studies RLHF scaling well and that the results do support the points in Section 4.4.

**Significance**: How well RLHF scales is likely of great interest to the broader ML community. RLHF/RLAIF have become extremely common place, and as it is more feasible now for non-commercial projects to do more intensive post-training I think this work is significant.

**Weaknesses:**

- The paper claims to study how RLHF scales, but they make some unconventional choices in how they design their RLHF pipeline. Notably, they use a single reward model for reasoning and human preference data. This weakens the results, as they do not directly assess RLHF as it is usually implemented.
- A very minor point: The paper mentions GRPO but never gives the expanded version of the acronym.

**Questions:**

- Would it be possible to run some smaller experiments without the unified reward model from section 3.1? If downstream policy model performance is similar even at smaller scales it would help show that your results track meaningfully to the case where separate reward models are used.
- Will you open source the reward models, corresponding policy models and the SFT model you use? I can see these models being useful for other work that studies how RLHF scales.

---

> ### Author Response · Authors · 2024-11-20
> **Response to Reviewer XHYP**
>
> Thanks for the reviewer’s response and appreciation of our work!
>
> **w1 & Q1: Using one reward model for both reasoning and human preference might weaken the results. More experiments on smaller models with separated reward models**
>
> Yes, we have also considered the issue of whether to use a unified reward model (RM) for all tasks or develop a separate RM for each individual task. Our findings indicate that employing a single unified RM achieves nearly the same performance as using a set of task-specific RMs in Best-of-N evaluations. Therefore, for the sake of training efficiency and simplicity, we opted to train a unified RM.
>
> We conducted experiments on mathematical tasks to assess this approach, and the Best-of-N results on the MATH dataset are as follows:
>
> |  | Best-of-4 (32B) | Best-of-16 (32B) | Best-of-4 (9B) | Best-of-16 (9B) |
> | --- | --- | --- | -- | -- |
> | math-only RM | 56.04 | 61.34 | 54.08 | 57.83 |
> | unified RM | 56.67 |  62.27 | 54.47 | 58.5 |
>
> As shown, the unified RM demonstrates performance comparable to the math-only RM and even exhibits a slight advantage, likely due to the inclusion of additional code reward model data in the unified RM. Overall, our results suggest that training a single reward model for multiple tasks is both feasible and does not compromise performance.
>
> **w2: The paper mentions GRPO but never gives the expanded version of the acronym.**
>
> Thanks for the kind suggestion. GRPO refers to Group Relative Policy Optimization and we have fixed the issue in the updated version (line 205).
>
> **Q2: will the reward and policy model be open-sourced?**
> Yes, we will open-source the reward model and also part of the training data. We hope that it could help the community to reproduce our results and contribute to further research in the field.

---

> > ### Comment · Reviewer_XHYP · 2024-11-25
> >
> > Thank you for the additional experiments and for fixing the minor issue with GRPO not being stated in full. Because my score is already quite high and representative of my views on the paper I will leave it unchanged.

---

### Official Review · Reviewer_AdSU · 2024-10-30

**Soundness:** 2
**Presentation:** 3
**Contribution:** 3
**Rating:** 6
**Confidence:** 4

**Summary:**

This work investigates the training scaling properties of RLHF for LLMs in the context of reasoning questions. They investigate two main settings: how does scaling effect the policy in RLHF assuming a fixed SFT model, and how does scaling the policy effect performance assuming a fixed RM and training strategy? They find that scaling up data, model size and training time often produces improvements, but these sometimes see diminishing returns at the high end of scaling up, even on a logarithmic x-axis. They additionally find that process supervision produces performance boosts over outcome supervision in-distribution but these improvements sometimes fail to generalise. Using these insights, the paper recommends practical ways in which increased compute can results in better performance for RLHF training for reasoning questions.

**Strengths:**

The paper performs extensive experiments across a range of scales and settings, making the results much more likely to be robust and generalisable. The topic is important and timely, and hasn't been investigated to this level of rigour before, making this a significant and original contribution. The paper is fairly well written and easy to understand. The research questions are well-scoped and investigated well. Overall, it makes a worthwhile contribution to our understanding of scaling properties in RLHF training.

**Weaknesses:**

# Paper framing

The paper title and introduction claims to address RLHF broadly construed, but the experimental setting is mostly focused on improvements in code and reasoning questions rather than more a more general chat setting. This is fine as a focus of the paper, but I think it would be beneficial to be clearer earlier in the paper that the RLHF setting considered is perhaps different from the standard one readers would expect (RLHF for dialogue).

# dataset and evaluation choice leads to lack of generality in conclusions

The paper uses a mix of datasets both for training and evaluation. However, it's unclear what the relationship between the training and evaluation datasets is, which means the results are harder to interpret. For example, when we see diminishing returns to scaling various properties, is that because these properties are not producing performance in-distribution in a clean manner, or because that in-distribution performance is not translating to the out-of-distribution evaluations being measured. In general, when measuring scaling trends like done in this paper, it's common practice to disentangle these two hypotheses by evaluating on in-distribution (but held out) data, but that is difficult in this setting given the heterogeneous nature of the RLHF training mixture. I believe the results in the paper are still interested and likely to be generalisable to some extent, but this experiment design decision does hamper the usefulness and transferability of the results to other settings. This is exemplified in the results in figure 2 - some benchmarks benefit from scaling of the properties investigated and some do not, but we don't know whether this is a generalisation failure or an optimisation failure, as we don't have in-distribution performance.

Additionally, it is difficult to calculate scaling trends for evaluation metrics such as those computed, as they're likely non-monotonic with respect to underlying metrics of performance. When observing that pretraining scaling predictably improves loss, this is easy as loss is grounded in the training procedure. However, evaluations based on metrics not directly optimised for means that it's difficult to explain diminishing returns to scale for that metric as scaling not working well, or whether that metric gets more difficult to improve the higher it is. Again, matching training and evaluation metrics and data more closely would address this problem.

This could be addressed firstly by making this limitation clear in the paper. It would also be beneficial to perform in-distribution evaluations of these models, where in-distribution means that both the input data and the reward function are matched to those that generated the training data for the policy and reward model respectively.

# contextualising results with respect to related work

Some of the key findings listed in the introduction are similar to those found in the literature. It would be beneficial to explicitly state where your results confirm previous findings, or disagree with them, or go beyond them.

# Unclear statements about comparison to pretraining scaling

In several places the paper claims that their results show that scaling RLHF is less effective that scaling pretraining. However, this comparison isn't made formal and hence I think this claim should be made more precise, or dropped from the paper. I don't think you can compare scaling in your setting (where training and evaluation objectives and data are different) to the pretraining scaling regime (where they are the same) without being clearer how this is done.

# Smaller issues

* One of your conclusions is that larger policy models benefit less from RLHF when using a fixed size reward model. However, this is confounded by the improved starting point of larger policy models, as the initial SFT is likely better. Combined with the issues above about the metric not being linear, this conclusion doesn't seem valid to me.
* You say "Recently, OpenAI-o1 (openai, 2024) has revealed the potential for scaling reinforcement learning at inference time and significantly boosts the reasoning abilities of LLMs." (line 135). However, o1 also scales RL at training time as well.
* when scaling responses per prompt, you're effectively scaling the batch size for training, but you're not also scaling the learning rate, which likely leads to worse performance than is achievable. In general larger batch sizes can accomodate larger learning rates and hence be more performant, and I think it would make more sense to adapt this hyperparameter to the setting to get more compelling results.
* It would be beneficial to have error bars or confidence intervals of some kind on most of the plots, to understand how noisy these results are. For example, in figure 2, MMLU and AlignBench move by neglible amounts, which could easily be noise in evaluation rather than a real trend.

# Summary

Overall, I think the paper is still worth of acceptance with several easy changes to writing and presentation, as described above. If those changes are made I would raise my score to a 6, and if more substantial experiments were done with in-distribution performance measures, and the smaller issues mentioned above were addressed, I would be happy to raise my score further.

# Update

I am happy to raise my score to a 6, assuming the clarifications and answers you offered in this response will be in the final version of the submission.

**Questions:**

It would be beneficial to get more details on the process supervision technique in the paper, so that it is somewhat self-contained, rather than just referencing another work without detailed explanation.

---

> ### Author Response · Authors · 2024-11-20
> **[Part 1/2] Response to Reviewer AdSU**
>
> Thanks for the reviewer’s comments and appreciation of our work!
>
> **w1: paper framing**
>
> Thanks for your suggestions. As the reviewer has suggested, we have added more explanations in the introduction part in the updated version (line 77-79). In this work, we mainly focus on reasoning-related tasks but also conduct evaluations on general tasks like AlignBench.
>
>
> **w2: Dataset and evaluation choice**
>
> >  it's unclear what the relationship between the training and evaluation datasets is, which means the results are harder to interpret.
>
> In our experiments, certain evaluation sets, such as MATH, GSM8k, and LiveCodeBench, are in-distribution relative to the training data, while others, such as GPQA, are not. And we see similar improvement trends in these benchmarks.
>
> Data distribution and the generalization of the trained model are indeed critical aspects of RLHF training. As described in the experiment section, our training data comprises the following components:
>
> - Mathematics data, including MATH-train, Numina-Math, and a Chinese K-12 dataset.
> - Code data, such as competition data from code-contest [1] and TACO [2].
> - General chat data, including ShareGPT and our self-collected open-domain chat data.
>
> We evaluate our model on six datasets: MATH, GSM8k, LiveCodeBench, MMLU, GPQA, and AlignBench. For reasoning-related benchmarks, MATH, GSM8k, and LiveCodeBench can be considered in-distribution evaluations, while GPQA serves as an out-of-distribution evaluation due to the absence of related data in our training set. As illustrated in Figure 2, these datasets demonstrate consistent scaling trends. Thus in-distribution and out-of-distribution might not be the direct key factors of scaling.
>
> Regarding MMLU and AlignBench, which do not exhibit clear scaling trends, it’s worth noting that MMLU performance is largely influenced by the pretraining phase and is less affected by RLHF. Similarly, human preference tasks tend to be less scalable compared to reasoning tasks.
>
> [1] Li Y, Choi D, Chung J, et al. Competition-level code generation with alphacode[J]. Science, 2022, 378(6624): 1092-1097.
>
> [2] Li R, Fu J, Zhang B W, et al. Taco: Topics in algorithmic code generation dataset[J]. arXiv preprint arXiv:2312.14852, 2023.
>
> > Downstream evaluation may not be a good metric to measure "scaling"
>
> Thanks for pointing this out! We had a similar thought process and tried to seek metrics that could better indicate the "scaling" in RLHF. Since the training loss indicates nothing in RLHF training, we have considered using reward on training or evaluation set. Unlike pre-training in which lower loss generally leads to better performance,  as shown in our paper,  higher reward does not indicate better performance. Therefore, we finally selected downstream evaluation metrics as the primary measure. Downstream evaluation is an appropriate choice, as it serves as a “golden reward,” providing a reliable signal to assess scaling improvements. As shown in OpenAI o1[1], they also use Pass@1 Accuracy in AIME to show the scaling trends in RLHF training and inference. Therefore, we are hoping this early attempt on this topic could spark more discussions and efforts in it.
>
> [1] https://openai.com/index/learning-to-reason-with-llms/
>
>
> **w3: Agreement / disagreement with previous related works**
>
> In the introduction, we list observations about RLHF training and scaling, some of which are first presented in this work and not proposed before:
>
> 1. Sampling more responses during training generally improves the policy model’s performance
> 2. Larger policy models benefit less from RLHF when using a fixed-size reward model
> 3. Performance improves remarkably in the early stage of training but the return gradually diminishes in later training.
>
> For the remaining ones, they go beyond or disagree with previous findings.
>
> 4. [Agree and Go beyond] Previous work also states that larger reward models (RM) show better performance in Best-of-N evaluation. In this work, we show that the larger RM can also benefit RLHF training but the improvement still significantly falls behind the gains in Best-of-N evaluation of the reward model.
> 5. [Agree and Go beyond] Previous work shows that increasing training responses of each prompt for RM improves its performance, but we show that increasing prompt diversity is more effective than increasing response diversity
>
> 6. [Go beyond] Previous work finds that process supervision might be better than outcome supervision.  But we show that PRM actually performs better in targeted tasks but struggles to generalize to other tasks.
>
> 7. [Disagree] About reward hacking, previous works show that RLHF could suffer from reward hacking and performance degradation, but we find that over-training in RLHF brings less performance improvement but does not result in degeneration.

---

> ### Author Response · Authors · 2024-11-20
> **[Part 2/2] Response to Reviewer AdSU**
>
> **w4: make a more clear statement about RLHF scaling and pretraining scaling**
>
> For pre-training scaling, the general definition is as follows: expanding model size, dataset and training compute in the training can lead to lower training loss and improve model performance.
>
> For RLHF scaling, there are more factors that could affect the final performance, with different policy model sizes, reward model sizes, and sampling budgets. And therefore, in the introduction part, we try to give a definition of RLHF scaling from two aspects:
> (1) Given a fixed SFT model, how does scaling other factors, including the reward model, sampling budgets, and training data, affect the policy model through RLHF?
> (2) With a fixed reward model and training strategy, whether a larger policy model can benefit more from RLHF?
>
> Through this expensive study, the main message we would like to share with the community is: Expanding reward model size (supervision signal), sampling budgets and training data can lead to better performance in RLHF,  but currently, larger policy models benefit less from RLHF when using a fixed-size reward model.
>
>
>
> **w5: Smaller issues**
>
> > add o1 RL training scaling and add error bar in figure 2
>
> Thanks for the kind suggestion. We have addressed the problem in the updated version.
>
> > why not using a larger learning rate for a larger batch size
>
> Yes, we have experimented with using a larger learning rate. However, we found that training becomes more prone to collapsing with higher learning rates, resulting in the model’s inability to generate proper responses. RLHF training is highly sensitive to the choice of learning rate: setting it too large can lead to model instability, while setting it too low can cause the learning to become too slow. Therefore, we used a learning rate of 2e-6 across all experiments to balance training stability and performance improvement. This learning rate works across different batch sizes.
>
> > why the starting point of larger models is lower in Figure 1(b)
>
> Figure 1(b) indicates the relative improvement of RLHF over the SFT policy. The result that the starting point of the larger model is lower indicates that the relative gain against the corresponding SFT policy from RLHF is lower in larger models than in smaller models. The result corresponds to conclusion 3 that "Larger policy models benefit less from RLHF when using a fixed size reward model"
>
>
>
> **Q1: more details on the process supervision technique**
>
> We applied the method by Math-Shepherd[1] to obtain an estimate of step correctness $Q(p,s)$ by running Monte Carlo rollouts for each step in the solution. Specifically, we divide a sampled solution $S$ of a problem $P$ into steps $S = S_1, S_2, ..., S_k$. For each step $S_i$, we perform $N$ rollouts to obtain $N$ generated solutions based on the same current step prefix $S_{i,j} = \\{S_{1,j}, ... S_{i, j}, S_{i+1, j}, ...,S_{k_j, j}\\}_j^N$, where $k_j$ is the total number of steps for the j-th finalized solution. We evaluate the correctness of the $N$ rollout results $A=\\{a_j\\}_j^N$. The correctness estimate of the current step is based on whether there is a correct solution among the $N$ rollouts at that step and it can be represented as $Q(P, S_i) =
> \begin{cases}
> 1 & \exists a_j \in A, a_j = 1 \\
> 0 & \text{Otherwise}
> \end{cases}$
> Then, we use the 0/1 binary classification labels of each step in solution $S$ as process supervision signals for PRM training.
>
> [1] Wang P, Li L, Shao Z, et al. Math-shepherd: Verify and reinforce llms step-by-step without human annotations[C]//Proceedings of the 62nd Annual Meeting of the Association for Computational Linguistics (Volume 1: Long Papers). 2024: 9426-9439.

---

> > ### Comment · Reviewer_AdSU · 2024-11-21
> >
> > Thanks for your response. I am happy to raise my score to a 6, assuming the clarifications and answers you offered in this response will be in the final version of the submission.
> >
> > While I appreciate your discussion of the other issues I raised, I don't think that remedies my concerns, and so I'm not planning on raising my score further without additional experiments.

---

> ### Author Response · Authors · 2024-11-25
> **Experiments on in-distribution performance evaluation**
>
> We conducted additional experiments to investigate whether data distribution impacts the scaling of RLHF. Specifically, we ensure that the reward function closely matched the policy model's output distribution, and that both the training set of RLHF and the evaluation dataset are under the same distribution, thus maintaining an in-distribution experimental setting.
> - For the reward model, as described in the experiment part (Line 232), all the training data is sampled from the SFT model and thus the reward function matches the distribution of generated data from the policy model for RLHF training.
> - For the RL training, to conduct the in-distribution experiment, we choose to use MATH-train[1] and MATH-test (the same as the MATH dataset in our paper) as the training and evaluation set to ensure they lie in the same distribution.
>
> The following table shows the results of RLHF training on different training dataset.
>
> |  Accuracy (%) | num_sample=1 | num_sample=4 | num_sample = 16 | Gain from 1-> 4 | Gain from 4->16 |
> | -- | -- | -- | -- | -- | -- |
> | Original training set | 50.44 | 52.28 | 52.76 | 1.84 | 0.48 |
> | MATH-train set | 50.52 | 53.56 | 54.64 | 3.04 | 1.08 |
>
> As shown in the table, training on the MATH-train dataset shows better performance than our original training set in MATH-test evaluation as the MATH-train and MATH-test are under the same distribution. However, they show similar trends that further increasing the sampled responses per prompt shows diminishing returns. Thus, the data distribution might not be the key factor in RLHF scaling.
>
> Thanks a lot for the valuable question and we also gain a lot from these experiments. Please let us know if you have any further questions, and we look forward to the possibility of your updated evaluation of our work.
>
>
> [1] Hendrycks D, Burns C, Kadavath S, et al. Measuring mathematical problem solving with the math dataset[J]. arXiv preprint arXiv:2103.03874, 2021.

---

### Official Review · Reviewer_Vekn · 2024-11-02

**Soundness:** 3
**Presentation:** 3
**Contribution:** 1
**Rating:** 3
**Confidence:** 4

**Summary:**

The paper proposes a sequence of experiments to show if the current RL recipe can scale.
The experiments range from reward modelling, testing different reward model sizes to generating more samples at training time or RL algorithm choice. They identify several problems with the current approach and conclude that scaling it is not feasible.

**Strengths:**

The paper showcases clearly the different shortcomings of the actual RLHF recipe to train LLMs.
The paper is a good technical report that reviews what are the different degrees of freedom in the mainstream RLHF recipe.
It explains that reward modelling is probably the main bottleneck towards scaling up RL methods.
The paper in its current state is more a technical report than a research paper in my opinion. My rating is based on this and not on the underlying quality of the document which is good.

**Weaknesses:**

My main concern with the paper is the lack of novelty and originality. There are no new findings obtained through the run experiments:
 - reward hacking is a known problem
 - the different RL approaches and reward normalization schemes are known
 - using N generations and how the performance plateaued is known

No solution is proposed to the main bottleneck which is reward modelling. If one wants RL to scale, it is also imperative to get rid of the anchor model as it constraints the optimal set of possible solutions. It is only used here to avoid the shortcomings of reward hacking. The authors should expand on this a little bit more. The authors do raise the point that increasing the reward value at training time does not correlate with improving performance with downstream tasks which shows that RLHF in its current state is not a proper training regime.
In addition, authors could have found potential directions of future research in the RL literature. To properly scale, especially in sparse environments RL methods need an exploration bonus or a way to understand their uncertainty about the environment. This is independent from a learnt reward model and could potentially scale. Authors should have at least tried to see if they could find a method to scale the diversity of outcomes in the obtained generations or if using different inference mechanisms (in addition to the sampling N parallel answers) could help scaling.

**Questions:**

I provided my list of suggestions in the previous section.

---

> ### Author Response · Authors · 2024-11-20
> **Response to Reviewer Vekn**
>
> We thank the reviewer for the comments and suggestions. We would like to further clarify the contribution of our work.
>
> 1. In this paper, we aim to systematically analyze the key factors that affect the scaling of RLHF and help the community better understand the scaling properties in RLHF training. Similar to our work, previous works [1, 2] investigate the scaling properties for synthetic reward modeling and supervised fine-tuning from the experimental perspective.  They also provide practical insights and contributions to research studies.
>
> 2. As noted by other reviewers, RLHF scaling has not been deeply investigated yet in the community. While some of the techniques we studied may have appeared in other literature, their scaling properties have not been thoroughly investigated in literature.
>
> There are indeed numerous factors that can be analyzed to understand their impact on the scalability of RLHF, including those discussed in our paper, such as model size and sampling budgets, as well as techniques mentioned in the reviewer’s comment, such as sampling strategies. In this work, our goal is to examine and differentiate between the scalable and non-scalable factors within the current RLHF framework. Specifically, the sampling strategies are not considered scalable factors to the RLHF training process. This provides a systematic understanding of the limitations and potential of the existing RLHF paradigm, in comparison to what has been extensively studied in scaling the pretraining. Based on the insights and findings, we aim to identify promising scaling directions and develop techniques to enhance the scalability of RLHF in the future.
>
> Therefore, we would like to articulate our contributions again in the following:
>
> 1. We identified key factors that have the potential to scale the impact of RLHF, including model size, data composition, reward signals, and sampling budget. And to study the scaling properties, we try to define and study the problem from two perspectives: fixed policy model while scaling other factors and scaling the size of the policy model.
> 2. We conduct systematic studies to understand the impact of these factors. These studies help us identify the limitations and scalable factors.
> 3. We found some conclusions that disagree with or go beyond previous works’ findings. For example, a larger policy model benefits less from RLHF training. Process supervision can lead to better performance in the targeted tasks but generalize worse.
>
> We aim to provide a better understanding for future research on RLHF scaling and offer actionable insights for practitioners seeking to further scale the training of RLHF.  For other techniques stated in the review, such as different sampling strategies,
>
>
> [1] Gao L, Schulman J, Hilton J. Scaling laws for reward model overoptimization[C]//International Conference on Machine Learning. PMLR, 2023: 10835-10866.
> [2] Zhang B, Liu Z, Cherry C, et al. When Scaling Meets LLM Finetuning: The Effect of Data, Model and Finetuning Method[C]//The Twelfth International Conference on Learning Representations.

---

> ### Author Response · Authors · 2024-11-22
> **Thanks for your kind review**
>
> We greatly appreciate your dedicated time and effort in reviewing our work. We kindly remind you to check if the points raised in your review have been addressed in my response. If any remaining concerns or areas require further clarification, we would be happy to provide additional details or explanations.

---

### Official Review · Reviewer_147F · 2024-11-03

**Soundness:** 2
**Presentation:** 2
**Contribution:** 2
**Rating:** 5
**Confidence:** 5

**Summary:**

This paper investigates key components in the RLHF framework, such as model size, data composition, and inference budget, assessing their scalability. The findings reveal that RLHF scales less efficiently than pretraining, with performance gains diminishing despite increased computational resources.

**Strengths:**

1.	This paper addresses a critical gap in current LLM post-training research by examining the scalability of RLHF.
2.	The experiments comprehensively cover various aspects of RLHF, including model size, data composition, and inference budget.
3.	The conclusions drawn are strongly supported by robust experimental results, providing clear insights into the limitations and potential of RLHF scalability.

**Weaknesses:**

1. RLHF encompasses a broad range of concepts, yet this paper does not cover all aspects of the literature. For instance, the impact of training data composition for the reward model on RLHF scalability is not explored.

2. While there are numerous RLHF approaches, such as DPO, RPO, and KTO, this paper focuses solely on PPO and GRPO. This limited scope challenges the claim of exploring the impact of methods comprehensively.

3. The study is primarily centered on reasoning tasks, such as math and coding, and does not extend to other important areas like general instruction-following tasks, which limits the generalizability of the findings.

4.  Discussion about potential hypotheses for why RLHF doesn't scale as well as pretraining and  experiments that could help isolate the cause are no presented.

**Questions:**

What is the reason that RLHF does not scale?
For instance, In Section 4.2.1, why the performance does not always improve when the number of responses increase?

---

> ### Author Response · Authors · 2024-11-20
> **[Part 1/2] Response to Reviewer 147F**
>
> Thanks for your kind and helpful suggestions!
>
> **w1: This paper does not cover all aspects of RLHF scaling.**
>
> Thanks for the suggestion! We agree that it is harder to define "scaling" in RLHF than that in pretraining scaling because there are much more factors that could affect the performance. Hence in the introduction part, we try to first give an overview of RLHF and then focus this study on the following aspects:
>
> 1. Given a fixed SFT model, how does scaling other factors affect the policy model through RLHF? In this aspect, we investigate the effects of reward model size from 9B to 200B, training data, sampling budget, and supervision signals.
> 2.  With a fixed reward model and training strategy, whether a larger policy model can benefit more from RLHF? In this aspect, we explore the gain from RLHF on different sizes of policy models from 9B to 200B and show our finding that larger policy models benefit less from RLHF when using a fixed-size reward model.
>
> We believe our early attempt on this topic will spark more discussions and efforts to better understand RLHF scaling.
>
> **w2: This paper focuses solely on PPO and GRPO and neglects other methods like DPO and KTO.**
>
> In this paper, we primarily investigate the scaling properties of on-policy RLHF methods, specifically PPO and GRPO. We updated the submission in the introduction to make it more clear (line 68-70).
>
> 1. Prior studies [1,2] demonstrate that PPO significantly outperforms DPO across a variety of tasks. Consequently, our analysis focuses on the scaling behavior of PPO and GRPO. Most works prefer DPO and KTO due to their simplicity, but PPO and GRPO exhibit better performance in downstream tasks.
> 2. Unlike PPO, DPO and KTO are off-policy methods. Previous work [3] has studied the scaling laws for off-policy approaches and showed that DPO-related methods could suffer from great over-optimization, and are not scalable.
>
> Therefore, this work mainly focuses on PPO and GRPO and explores their potential for scaling.
>
> [1] Ivison H, Wang Y, Liu J, et al. Unpacking DPO and PPO: Disentangling Best Practices for Learning from Preference Feedback[J]. arXiv preprint arXiv:2406.09279, 2024.
>
> [2] Shao Z, Wang P, Zhu Q, et al. Deepseekmath: Pushing the limits of mathematical reasoning in open language models[J]. arXiv preprint arXiv:2402.03300, 2024.
>
> [3] Rafailov R, Chittepu Y, Park R, et al. Scaling laws for reward model overoptimization in direct alignment algorithms[J]. arXiv preprint arXiv:2406.02900, 2024.
>
> **w3: The study is primarily centered on reasoning tasks and does not extend to other important areas like general instruction-following tasks**
>
> In this work, we focus primarily on reasoning-related tasks as previous works have shown that post-training shows scaling potential in RLHF tasks[1, 2].  But we also conduct experiments on general instruction-following tasks, as outlined in Section 4.1.
> - For training, we curate a dataset comprising both general chat and reasoning data to train the reward model and the policy model.
> - For evaluation, we assess our model using AlignBench, a widely recognized benchmark for evaluating the general alignment of large language models (LLMs), and MMLU. The results are presented in Figure 2 and we put the results on AlignBench again as follows:
>
> AlignBench measures performance on general instruction-following and chatting tasks, as well as the effectiveness of Reinforcement Learning with Human Feedback (RLHF) on human preference tasks.
>
> As observed, while RLHF improves performance on these tasks, scaling, including larger reward models, or more sampled responses, does not yield significant benefits for general instruction-following tasks, unlike reasoning tasks.
>
> | num_responses | 1 | 2 | 4 | 8 | 16 |
> | ---- | --- | --- | --- | --- | --- |
> | Reward-9B | 7.44 | 7.45 | 7.59 | 7.59 | 7.41 |
> | Reward-32B | 7.47 | 7.42 | 7.46 | 7.58 | 7.45 |
>
> [1]  https://openai.com/index/learning-to-reason-with-llms/
>
> [2] Yuan Z, Yuan H, Li C, et al. Scaling relationship on learning mathematical reasoning with large language models[J]. arXiv preprint arXiv:2308.01825, 2023.

---

> ### Author Response · Authors · 2024-11-20
> **[Part 2/2] Response to Reviewer 147F**
>
> **w4: Discussion about potential hypotheses for why RLHF doesn't scale as well as pretraining and experiments that could help isolate the cause is not presented.**
>
> In the discussion section (4.4), we discuss the limitations and potential issues in the current RLHF.  We conclude that the potential challenges hindering the scalability of RLHF can be attributed to two key factors: inaccuracies in reward signals and the inherent difficulty of training in RLHF. These challenges are evident in two types of gaps: the RLHF improvement largely lags behind the Best-of-N results, and the Best-of-N results largely lag behind Pass@K , as outlined in the following tables.
>
>
> In comparison, pretraining relies on next-token prediction as the supervision signal, which comes from the text data itself, and also involves abundant of tasks. That might be the reason why the data quality is supreme in pretraining because it determines the quality of supervision signals.
>
> As shown in our experiments, more accurate reward signals(i.e., a larger reward model) can lead to better scaling trends in RLHF.
> Larger reward models demonstrate significantly improved performance in both best-of-N evaluation and RLHF results on reasoning tasks, as illustrated in Figure 1. Thus more precise reward signals can enhance the scalability of RLHF training. However, the performance of current reward models remains suboptimal, as the Best-of-N results fall considerably short of Pass@N (Correctness of golden answer as the reward). We propose that the inaccuracy of reward signals may be a limiting factor and that refining these signals could play a crucial role in advancing RLHF scalability.
>
> RLHF results
> | Reward Model | 9B | 32B | 200B |
> | --- | --- | --- | --- |
> | MATH | 51.44 | 53.52 | 54.24 |
> | Code | 23.5 | 24.75 | 27.25 |
> | GPQA | 30.1 | 32.63 | 33.23 |
>
> Best-of-8 results:
> | Reward Model | 9B | 32B | 200B | Gold Answer |
> | --- | --- | --- | --- | --- |
> | MATH | 56.37 | 59.19 | 62.42 |  78.20 |
> | Code | 21.57 | 23.99 | 27.4 | 29.00 |
> | GPQA | 32.48 | 34.62 | 37.44 | 73.74 |
>
> In addition, we also find some factors that might help RLHF scaling in the future, like sampling multiple responses, and process supervision.

---

> ### Author Response · Authors · 2024-11-22
> **Thanks for your kind review**
>
> We greatly appreciate your dedicated time and effort in reviewing our work. We kindly remind you to check if the points raised in your review have been addressed in my response. If any remaining concerns or areas require further clarification, we would be happy to provide additional details or explanations.

---

> > ### Comment · Reviewer_147F · 2024-11-27
> >
> > Based on the last part of the discussion, more precise reward signals can enhance the scalability of RLHF training. Then for certain tasks such as coding or math problems, we can use test case or ground truth answer to get a fairly accurate reward. In such cases, whether RLHF scales?

---

> ### Author Response · Authors · 2024-11-25
>
> Thanks for your efforts in reviewing the paper and your valuable question! We hope that our responses could address your concerns. We also conduct additional experiments to investigate the effects of data distribution in the reward model and RLHF training.
>
> 1. We compare the performance of training a unified reward model or a specific reward model for each task. We conducted experiments on mathematical tasks to assess this approach, and the Best-of-N results on the MATH dataset are as follows:
>
> |  | Best-of-4 (32B) | Best-of-16 (32B) | Best-of-4 (9B) | Best-of-16 (9B) |
> | --- | --- | --- | -- | -- |
> | math-only RM | 56.04 | 61.34 | 54.08 | 57.83 |
> | unified RM | 56.67 |  62.27 | 54.47 | 58.5 |
>
>
> 2. We conduct experiments to test whether the data distribution could affect the scaling of RLHF.  We ensure that the reward function closely matches the policy model's output distribution and that both the training set of RLHF and the evaluation dataset are under the same distribution, thus maintaining an in-distribution experimental setting. Specifically, we conduct RLHF training on MATH-train only and evaluate the performance on MATH-test (the same as the MATH dataset in our paper) as the training and evaluation set to ensure they lie in the same distribution.
>
> |  | num_sample=1 | num_sample=4 | num_sample = 16 | Gain from 1-> 4 samples | Gain from 4->16 samples |
> | -- | -- | -- | -- | -- | -- |
> | Original training set | 50.44 | 52.28 | 52.76 | 1.84 | 0.48 |
> | MATH-train set | 50.52 | 53.56 | 54.64 | 3.04 | 1.08 |
>
> We hope that these experiments can help you better evaluate our work. Please let us know if you have any further questions, and we look forward to the possibility of your updated evaluation of our work.

---

> ### Author Response · Authors · 2024-11-30
>
> Thanks for your response! As shown in the results above, more precise reward signals can improve performance in RLHF training. However, we observe diminishing returns as we increase the number of sampled responses per prompt from 4 to 16, compared to the increase from 1 to 4. As noted in [1], they found that using ground truth rewards for code/math tasks can sometimes lead to worse performance than using reward models. This is because, for example, in coding tasks, limited test cases may not offer full coverage, making the binary feedback (0-1) noisy and suboptimal. Reward models, in contrast, are more robust and have better generalization capabilities.
>
> Additionally, we are currently conducting experiments using ground truth rewards. Since we need to modify our implementation to enable this training, we will share the results as soon as we have them.
>
>
> [1] Zhu Q, Guo D, Shao Z, et al. DeepSeek-Coder-V2: Breaking the Barrier of Closed-Source Models in Code Intelligence[J]. arXiv preprint arXiv:2406.11931, 2024.

---

> ### Author Response · Authors · 2024-12-01
> **Compare reward model and groud truth reward**
>
> We conduct experiments to compare the performance using reward model or groudtruth as reward in RLHF training.  We use MATH-train as the training set and MATH-test (the same as the MATH dataset in our paper) for evaluation set. The results are as follows:
>
> |  | num_sample=1 | num_sample=4 | num_sample = 16 | Gain from 1-> 4 samples | Gain from 4->16 samples |
> | -- | -- | -- | -- | -- | -- |
> | Reward Model(9B) | 50.52 | 53.56 | 54.64 | 3.04 | 1.08 |
> | GroundTruth Label | 49.68 | 52.44 | 53.54 | 2.76 | 1.1 |
>
> As observed, both using the reward model and the ground truth as reward signals show similar trends, with the reward model demonstrating slightly better performance. This aligns with the earlier discussion. The reason for this could be that signals from the reward model are continuous and more robust. However, we also noticed an interesting trend in the experiments where 16 responses were sampled per prompt: when using the reward model, performance peaks around 1/2 to 2/3 of the total training steps (approximately 70 steps). In contrast, when using ground truth as the reward signal, performance continues to improve even after one epoch of training. This suggests that, with more math-specific training data and by sampling more responses per prompt, training with ground truth may offer better scalability. Due to time and resource constraints, we were only able to conduct experiments under the current settings. We will continue our exploration and hope to report further results in the next version of our paper.
>
> We hope that these experiments can address your concerns and questions. Please let us know if you have any further questions, and we look forward to the opportunity to receive your updated evaluation of our work

---

### Meta-Review · Area_Chair_XeZx · 2024-12-23

**Metareview:**

This paper performs a systematic study to understand scaling properties of RLHF algorithms on reasoning tasks. While the paper presents some interesting conclusions and studies, the reviews were mixed between accept and reject scores. At a high level, the AC agrees with some of the reviewers that the conclusions in this paper, while interesting, are not super rigorous along any one dimension. It seems like the paper prioritizes coverage of multiple hypothesis, but leaves some questions open along each of them.

For example, while the title indicates that the paper studies scaling of RLHF more generally, the only domains are reasoning based and not general instruction tuning; while the paper studies PRMs, the only PRMs are trained based on Math-Shepherd data collection schemes; observations about large policy and reward models are to a large extent known in literature (cf scaling laws of reward overoptimization paper); comparisons to pre-training scaling laws are not defined rigorously enough; RLOO and DPO style RL algorithms are not looked upon (though they form a big chunk of algorithms that some in RLHF community use).

I would suggest that the authors take into account some of the reviewer suggestions and make the paper solid along some axes, and avoid the temptation of studying many axes but not spending enough time on any one. Unfortunately, we are not able to accept the paper right now due to these reasons.

**Additional Comments On Reviewer Discussion:**

The main points raised by the reviewers largely fall into the category of digging deeper into some aspects that I agree with. Some of the reviewers didn't respond to the rebuttal, but the decision does take into account the author responses in that case.

---

### Decision · Program_Chairs · 2025-01-22

Reject